# Selective Adsorption Behavior and Mechanism for Cd(II) in Aqueous Solution with a Recoverable Magnetie-Surface Ion-Imprinted Polymer

**DOI:** 10.3390/polym15112416

**Published:** 2023-05-23

**Authors:** Siqing Ye, Weiye Zhang, Xingliang Hu, Hongxing He, Yi Zhang, Weili Li, Guangyuan Hu, Yue Li, Xiujun Deng

**Affiliations:** 1Yunnan Key Laboratory of Food Safety Testing Technology, College of Chemistry and Chemical Engineering, Kunming University, Kunming 650214, China; 18860965309@163.com (S.Y.); zhangviye@icloud.com (W.Z.);; 2Kunming Lüdao Environmental Technology Co., Ltd., Kunming 650228, China; 3Yunnan Key Laboratory of Metal-Organic Molecular Materials and Device, School of Chemistry and Chemical Engineering, Kunming University, Kunming 650214, China

**Keywords:** ion-imprinted polymer, magnetic separation, Cd(II) ion, selective adsorption, density functional theory

## Abstract

A novel recoverable magnetic Cd(II) ion-imprinted polymer was synthesized on the surface of silica-coated Fe_3_O_4_ particles via the surface imprinting technique and chemical grafting method. The resulting polymer was used as a highly efficient adsorbent for the removal of Cd(II) ions from aqueous solutions. The adsorption experiments revealed that Fe_3_O_4_@SiO_2_@IIP had a maximum adsorption capacity of up to 29.82 mg·g^−1^ for Cd(II) at an optimal pH of 6, with the adsorption equilibrium achieved within 20 min. The adsorption process followed the pseudo-second-order kinetic model and the Langmuir isotherm adsorption model. Thermodynamic studies showed that the adsorption of Cd(II) on the imprinted polymer was spontaneous and entropy-increasing. Furthermore, the Fe_3_O_4_@SiO_2_@IIP could rapidly achieve solid–liquid separation in the presence of an external magnetic field. More importantly, despite the poor affinity of the functional groups constructed on the polymer surface for Cd(II), we improved the specific selectivity of the imprinted adsorbent for Cd(II) through surface imprinting technology. The selective adsorption mechanism was verified by XPS and DFT theoretical calculations.

## 1. Introduction

Cadmium is considered a highly toxic metal with high bioaccumulation potential and migratory capacity in water, and its presence can adversely affect human health (as a carcinogen and teratogen causing damage to the kidneys and brain) and aquatic ecosystems (by inhibiting physiological processes) [1]. Anthropogenic activities, such as metallurgical industries, mining operations, agricultural sectors, and paint industries, are the primary sources of increased cadmium concentrations in water [2]. The World Health Organization and the Ministry of Ecology and Environment of the PRC have established upper limits for Cd(II) concentrations in drinking water at 0.003 mg·L^−1^ and 0.005 mg·L^−1^, respectively, to prevent adverse effects on human health and aquatic ecosystems [3,4]. Consequently, the removal of Cd(II) from aqueous solutions is of great significance, and efficient techniques for this purpose are highly sought after. Adsorption is a cost-effective and efficient method for removing Cd(II) from environmental water systems, but traditional adsorbents lack the ability to selectively separate target metal ions, thus limiting their application [5].

Ion-imprinted polymers (IIPs) are highly desirable due to their selectivity for target ions in the presence of competing ions. They are a type of biomimetic polymer material prepared using imprinted ions as templates and possessing a specific memory and the ability to recognize imprinted ions [6,7]. IIPs have several advantages, including specific recognition, predictable conformation, and easy operation. However, traditional IIPs suffer from the imprinted sites embedded within the polymer network having poor accessibility, which results in low mass transfer rates for target ions [8,9]. Surface ion-imprinting technology (SIIT) has been developed to combat these limitations [10]. The technique consists of grafting imprinting sites onto the matrix’s surface, improving the speed and ease of the target metal ion’s binding to the imprinting sites. Surface ion-imprinted polymers (SIIPs) synthesized with SIIT exhibit advantageous properties, such as high selectivity, more accessible sites, high adsorption capacity, and fast mass transfer rates [11]. Montmorillonite [12], mesoporous silica [13], multi-walled carbon nanotubes [14], and Fe_3_O_4_ nanoparticles [15] have been widely used as matrix materials. Fe_3_O_4_ nanoparticles possess superparamagnetism and a high specific surface area and their surfaces can be easily modified, enabling rapid separation under the action of an external magnetic field. They effectively solve the solid–liquid separation problems affecting imprinted materials in solution, enhance the efficiency of separation and recovery, and hold promising development prospects [16].

2-Phosphonobutane-1,2,4-tricarboxylic acid (PBTCA) is an environmentally friendly, phosphonic acid-based surfactant that contains both carboxyl and phosphonic groups in a single molecule. PBTCA has been shown to improve the coordination of heavy metal ions [17]. Huang et al. developed a novel adsorbent by modifying chitosan-coated magnetic silica nanoparticles with PBTCA and demonstrated its ability to selectively adsorb uranium from aqueous solutions [18]. Cao et al. reported that adding PBTCA to plant agents significantly improved the capabilities for removal of heavy metals, such as Cd(II), Pb(II), and Zn(II), from contaminated soil [19]. To date, however, there is no research documenting the use of surface-imprinted magnetic polymers that employ PBTCA as an imprint site to selectively remove Cd(II) from aqueous solutions.

In this study, a novel magnetic imprinted polymer was developed utilizing SIIT with Cd(II) as the template molecule, Fe_3_O_4_ as the magnetic core, and PBTCA as the functional monomer. Chemical grafting and imprinting techniques were employed to introduce carboxyl and phosphonic acid functional groups onto the surface of Fe_3_O_4_ nanoparticles, resulting in a selective adsorbent for Cd(II) removal from aqueous solutions. Batch adsorption experiments were conducted to evaluate the adsorption performance of the imprinted adsorbent for Cd(II). The samples were characterized with several techniques, including FTIR, TEM, EDS, VSM, and TGA. Furthermore, XPS and DFT calculations were utilized to analyze the selective adsorption mechanism of the adsorbent for Cd(II).

## 2. Materials and Methods

### 2.1. Reagents

HCl and NaOH were purchased from Kolon Chemical Co., Ltd. (Chengdu, China). Ni(NO_3_)_2_·6H_2_O, Co(NO_3_)_2_·7H_2_O, Zn(NO_3_)_2_·7H_2_O, and Cd(NO_3_)_2_·4H_2_O were purchased from Beijing Chemical Plant (Beijing, China). Ammonium persulfate and sodium bisulfate were purchased from Windship Chemical Reagent Technology Co. (Tianjin, China). *N,N*’-Methylenebisacrylamide (MBA) was purchased from Maclean Biochemical Technology Co. (Shanghai, China). 2-Phosphonobutane-1,2,4-tricarboxylic acid (PBTCA) was purchased from Aladdin Biochemical Technology Co. (Shanghai, China). All aqueous solutions were prepared with deionized water.

### 2.2. Synthesis of Fe_3_O_4_@SiO_2_@IIP

The core-shell magnetic Fe_3_O_4_@SiO_2_@IIP nanospheres were obtained using surface ion-imprinting technology and the chemical grafting method, and the synthesis procedure is shown in Figure 1.

First, the vinyl-modified magnetic matrix material Fe_3_O_4_@SiO_2_@VTMOS was prepared according to our previous report [20]. Second, the Fe_3_O_4_@SiO_2_@IIP nanospheres were prepared with the following steps. Briefly, 3 mmol Cd(NO_3_)_2_·4H_2_O and 1 mmol PBTCA were added to 50 mL deionized water and stirred magnetically at room temperature. After stirring for 0.5 h, 0.15 g Fe_3_O_4_@SiO_2_@VTMOS was added and stirred for 12 h. Then, 8 mmol MBA, 0.1 g ammonium persulfate, and 0.1 g sodium bisulfite were added, bubbled with high-purity argon for 30 min to eliminate oxygen, and sealed. After that, the mixture was heated in a water bath at 50 °C and magnetically stirred for 5 h. Finally, the obtained polymers were separated with a magnet; washed with methanol, hydrochloric acid, and deionized water to neutrality; and dried in a vacuum at 40 °C for 12 h to obtain Fe_3_O_4_@SiO_2_@IIP.

The synthesis of the Fe_3_O_4_@SiO_2_@NIP adsorbent was roughly the same as the above steps, except that Cd(NO_3_)_2_·4H_2_O was not introduced and HCl was not used for washing.

### 2.3. Adsorption Experiments

Batch adsorption experiments were performed by adding a certain amount of sorbent to a certain volume of metal ion solution. The mixture was shaken for a certain time at 25 °C. The concentrations of all metal ions were determined with ICP-AES. Each adsorption experiment was conducted in triplicate. The adsorption capacity was calculated according to Equation (1) [21]:(1)$$q=\frac{c_{i}-c_{f}}{1000W}\times V$$
where q (mg·g^−1^) is the adsorption capacity; $c_{i}$ (mg·L^−1^) and $c_{f}$ (mg·L^−1^) are the initial and final concentrations of metal ions, respectively; V (mL) is the volume of the solution; and W (g) is the amount of sorbent.

### 2.4. Selectivity Study

In the selective adsorption experiment, 10 mg of the adsorbents Fe_3_O_4_@SiO_2_@IIP or NIP was added to 20 mL of a 5 mg·L^−1^ mixture of Cd(II), Ni(II), Co(II), and Zn(II) at solution pH = 6 and shaken for 1 h. Finally, the ion concentration in the filtrate was detected with ICP-AES. The distribution ratio (*D*), the selectivity coefficient (*k*) and the relative selectivity coefficient (*k′*) were calculated according to Equations (2)–(4) [22]:(2)$$D=\frac{\left(c_{0}-c_{e}\right)}{c_{e}}\times \frac{V}{W}$$
(3)$$k=\frac{D_{Cd}}{D_{M}}$$
(4)$$k^{\prime }=\frac{k_{IIP}}{k_{NIP}}$$
where $c_{0}$ (mg·L^−1^) and $c_{e}$ (mg·L^−1^) are the initial and equilibrium concentrations of metal ions and *M* represents other competitive metal ions.

### 2.5. DFT Calculations

DFT calculations were performed using Gaussian 16W to obtain the adsorption energy. The system was optimized in aqueous solution using a PBE1PBE/def2svp basis set and D3(BJ) dispersion energy correction. After convergence, the def2tzvp basis set was used to calculate the adsorption energy ($E_{ad}$). The adsorption energy ($E_{ad}$) between the PTBCA molecule and different metal ions was calculated using Equation (5) [23]:(5)$$E_{ad}=E_{\left(total\right)}-E_{\left(PBTCA\right)}-E_{\left(absorbates\right)}$$
where $E_{\left(total\right)}$, $E_{\left(PBTCA\right)}$, $E_{\left(absorbates\right)}$ are the total energy of the adsorption complex, the PTBCA molecule and the adsorbates, respectively.

## 3. Results

### 3.1. Characterization

#### 3.1.1. TEM

Figure 2 shows the TEM images of Fe_3_O_4_@SiO_2_ and Fe_3_O_4_@SiO_2_@IIP. The sequentially coated SiO_2_ and PBTCA imprinted layers still maintained the spherical structure with good dispersion and did not appear to be clustered. In addition, the nucleus in the dark part represents Fe_3_O_4_ nanoparticles, the inner layer with medium contrast is the silica shell, and the outer layer with light contrast is the PBTCA imprinted layer [22]. Figure 2(a1) confirms the successful modification of Fe_3_O_4_ nanoparticles through the silica layer with a particle size of about 519.1 nm, and Figure 2(b1) shows a significant increase in the thickness of the magnetic-carrier cladding layer with a particle size of about 783.7 nm. In summary, the results showed that Fe_3_O_4_@SiO_2_@IIP was prepared successfully.

#### 3.1.2. EDS

The EDS spectra of Fe_3_O_4_@SiO_2_@IIP before and after the adsorption of Cd(II) ions are shown in Figure 3. The presence of Fe and Si elements indicates the successful modification of Fe_3_O_4_ nanoparticles through the silica layer. The Cd changed significantly before and after the adsorption, with the content shifting from 0.75% (Wt%) to 6.36% (Wt%), confirming the adsorption properties of Fe_3_O_4_@SiO_2_@IIP.

#### 3.1.3. XRD

Phase identification was undertaken using XRD analysis. XRD patterns of the as-prepared Fe_3_O_4_, Fe_3_O_4_@SiO_2_, Fe_3_O_4_@SiO_2_@IIP, and Fe_3_O_4_@SiO_2_@NIP are shown in Figure 4. Compared with Fe_3_O_4_, Fe_3_O_4_@SiO_2_ showed the diffraction peaks of Fe_3_O_4_ at angles of 30.0°, 35.3°, 42.7°, 53.1°, 56.7°, and 62.3° corresponding to the (220), (311), (400), (422), (511), and (440) crystal planes of Fe_3_O_4_, respectively [24,25], which confirmed the successful compounding of Fe_3_O_4_ microspheres with SiO_2_. The XRD diffraction peaks of Fe_3_O_4_@SiO_2_@IIP and Fe_3_O_4_@SiO_2_@NIP were almost identical and became cluttered but, again, both retained the crystal plane of Fe_3_O_4_, verifying the successful introduction of Fe_3_O_4_ into the system of imprinted polymers. The detection of a characteristic peak at 35.3° confirmed the presence of high-crystalline Fe_3_O_4_ in the four samples.

#### 3.1.4. Surface Area and Pore Size Analysis

The adsorption mechanism of the imprinted polymers was further studied in a N_2_ adsorption–desorption experiment. The adsorption and desorption isotherms and the surface physical parameters are shown in Figure 5 and Table 1. The surface area was analyzed with the Brunauer–Emmett–Teller (BET) method, and the pore size distributions were determined with the Barrett–Joyner–Halenda (BJH) method. It can be seen from the figure that the shapes of the adsorption/desorption isotherms for Fe_3_O_4_@SiO_2_@IIP could be fitted to the classic II isotherm in the IUPAC classification method, which indicates the typical characteristics of microporous materials, showing that there were two kinds of adsorption sites. The surface area and pore volume of Fe_3_O_4_@SiO_2_@IIP were 107.990 m^2^·g^−1^ and 0.235 cm^3^·g^−1^, respectively, higher than those of Fe_3_O_4_@SiO_2_@NIP (79.455 m^2^·g^−1^ and 0.182 cm^3^·g^−1^), which could be attributed to the specific recognition cavities for Cd(II) formed on the surface of the sorbent by the imprinting technique [26].

#### 3.1.5. FTIR Spectroscopic Analysis

The FTIR maps of the samples were swept at 4000~400 cm^−1^ using the KBr infrared spectroscopy method to analyze the structures of the materials. Figure 6 shows the infrared spectra of Fe_3_O_4_ (a), Fe_3_O_4_@SiO_2_ (b), Fe_3_O_4_@SiO_2_@VTMOS (c), and Fe_3_O_4_@SiO_2_@IIP (d). A strong adsorption peak at 582 cm^−1^ (a) was observed for the stretching vibration of the Fe–O band in Fe_3_O_4_ magnetic nanoparticles [27]. The peaks for Fe-O at about 582 cm^−1^ in all the FTIR spectra (Figure 6a–d) confirmed the presence of Fe_3_O_4_ in all the nanospheres. After TEOS treatment of the Fe_3_O_4_, two peaks appeared at 1630 cm^−1^ and 1083 cm^−1^ (b). The former was due to the existence of O–H and the latter was the stretching vibration of Si–O, which verified that the Fe_3_O_4_ nanospheres were coated with SiO_2_ shells [28]. The peak at 1629 cm^−1^ (c) was caused by C=C stretching vibration. This proved that the double bond was successfully grafted onto the surface of Fe_3_O_4_@SiO_2_ [29].

The new peaks appearing at 1650 and 1203 cm^−1^ (d) were assigned to the characteristic peaks of C=O and P=O bonds [30], respectively. This confirmed the abundant presence of carboxyl groups and phosphonic groups on the Fe_3_O_4_@SiO_2_@IIP surfaces. In addition, the characteristic peak of the NH bond at 1527 cm^−1^ revealed the presence of MBA as crosslinker [31]. The appearance of these peaks indicated that the PBTCA was successfully grafted onto the Fe_3_O_4_@SiO_2_@VTMOS surface. These results demonstrated that Fe_3_O_4_@SiO_2_@IIP was successfully synthesized via surface-imprinting polymerization.

#### 3.1.6. Magnetic Separation Performance

As shown in Figure 7, the hysteresis regression curves of the three materials had similar shapes and all passed through the origin, indicating their superparamagnetic nature and strong magnetic responsiveness to magnetic fields. This responsiveness is beneficial for magnetic separation purposes. The saturation magnetization intensity of Fe_3_O_4_ at room temperature was found to be 66.79 emu/g, while that of Fe_3_O_4_@SiO_2_ was reduced to 63.89 emu/g. The observed reduction in saturation magnetization intensity could be attributed to the presence of a SiO_2_ layer on the surface of the Fe_3_O_4_ nanoparticles. This layer resulted in a decrease in the mass fraction of the magnetic components. The saturation magnetization intensity of Fe_3_O_4_@SiO_2_@IIP was found to be significantly lower than that of bare Fe_3_O_4_ (15.02 emu/g). This reduction in saturation magnetization intensity can be attributed to the shielding effect of the polymer shell on the surface of the Fe_3_O_4_ [27]. As shown in the inset image in Figure 5, Fe_3_O_4_@SiO_2_@IIP could be completely separated from the solution by an external magnetic field. The black dispersions after magnetic separation could become clear and transparent in a very short time (about 25 s). These results confirmed that the obtained Fe_3_O_4_@SiO_2_@IIP had outstanding magnetic properties and could be used for fast magnetic separation.

#### 3.1.7. Thermal Stability Analysis

The thermal stability of the magnetic Cd(II) ion-imprinted material was studied with TGA in the range of 25 °C~800 °C at a heating rate of 10 °C min^−1^ under a nitrogen atmosphere. Fe_3_O_4_ (Figure 8a) and Fe_3_O_4_@SiO_2_ (Figure 8b) underwent a first drop at 150 °C, but it was not significant and probably due to the evaporation of water. Fe_3_O_4_ and Fe_3_O_4_@SiO_2_ demonstrated weight losses of 23.5% and 8.22%, respectively, as the temperature increased from 25 °C to 800 °C. This was attributed to the successful coating of SiO_2_ on the surface of the Fe_3_O_4_. The modification of SiO_2_ resulted in an improvement in the thermal stability of Fe_3_O_4_@SiO_2_. For Fe_3_O_4_@SiO_2_@IIP (Figure 8c), the first weight loss was about 17.1% and occurred as the temperature changed from room temperature to 150 °C; it was attributed to the loss of the adsorbed water in the polymers, which was not eliminated during the drying process. When the temperature increased from 150 to 260 °C, the thermal weight loss was minimal, indicating that the imprinted material could remain stable below 260 °C. However, the weight loss was significant in the temperature range from 260 to 480 °C at about 45.9%, which may have been due to the decomposition of the imprinted polymer layer on the surface of the Fe_3_O_4_@SiO_2_. The results showed that the prepared magnetic Cd(II) imprinted material had good thermal stability at temperatures below 260 °C.

### 3.2. Effect of pH on Adsorption

The pH of a solution has a significant impact on the adsorption of metal ions as it affects both the morphology of the ions in the aqueous solution and the charge distribution on the surface of the adsorbent material [32]. Given that Cd(II) ions hydrolyze when pH > 8.0, the effects of different pH values on the adsorption properties of Cd(II) were studied at room temperature under the conditions of a Cd(II) concentration of 50 mg·L^−1^ and pH values in the range of 2.0–8.0. The results are shown in Figure 9. The adsorption capacity of the Fe_3_O_4_@SiO_2_@IIP for Cd(II) increased gradually with the increase in solution pH and reached the maximum value of 13.88 mg·g^−1^ at the solution pH = 6.0. This may have been due to the high protonation of the imprinted polymer surface groups in the form of -COOH at low pH conditions, which caused severe repulsion from Cd(II) in solution. As the pH rose, the degree of protonation on the adsorbent surface gradually decreased, making it easier for Cd(II) ions to bind with carboxyl and phosphate groups on the surface of the adsorbent, thereby enhancing the chelation of Cd(II). When pH > 6.0, the adsorption capacity decreased with the increasing pH. This may have been due to the fact that it was Cd(II) that was hydrolyzed and precipitated, and some of the Cd(II) in the system was hydrolyzed to Cd(OH)^+^, reducing the repulsive forces between them and resulting in a lower concentration of free Cd(II) and, thus, lower adsorption [33,34]. Therefore, a pH value of 6.0 in aqueous solution was selected as the optimal pH for the subsequent experiments.

### 3.3. Adsorption Kinetics

The adsorption kinetics for Cd(II) were assessed by measuring the adsorption equilibrium time for the adsorbent under conditions of an initial Cd(II) concentration of 50 mg·L^−1^, pH = 6.0, and a temperature of 25 °C. As shown in Figure 10, Fe_3_O_4_@SiO_2_@NIP obtained adsorption equilibrium faster than Fe_3_O_4_@SiO_2_@IIP but with a reduced adsorption capacity. Fe_3_O_4_@SiO_2_@NIP had a lower uptake rate due to the smaller surface area and reached equilibrium earlier compared to Fe_3_O_4_@SiO_2_@IIP. Owing to the presence of template Cd(II) ions, Fe_3_O_4_@SiO_2_@IIP had more adsorption sites than Fe_3_O_4_@SiO_2_@NIP, resulting in a higher adsorption capacity (18.21 mg·g^−1^) and a longer time (22 min) being required to reach adsorption saturation.

To further investigate the kinetics of the adsorption mechanism of the imprinted materials, pseudo–first–order kinetics (Equation (6)), pseudo–second–order kinetics (Equation (7)), and Weber–Morris kinetic models (Equation (8)) were used for linear fitting of the data [35]:(6)$$\ln\left(q_{e}-q_{t}\right)=lnq_{e}-k_{1}t$$
(7)$$\frac{t}{q_{t}}=\frac{1}{k_{2}q_{e}^{2}}+\frac{1}{q_{e}}t$$
(8)$$q_{t}=K_{i}t^{0.5}+C$$
where $q_{t}$ (mg·g^−1^) and $q_{e}$ (mg·g^−1^) are the adsorption capacities of Cd(II) at time t (min) and equilibrium, respectively; $k_{1}$ (min^−1^) is the rate constant of the first-order model; $k_{2}$ (g·min^−1^·mg^−1^) is the rate constant for the pseudo-second-order model at equilibrium; $K_{i}$ (mg·g^−1^·min^0.5^) indicates the rate constants for the Weber intra-particle diffusion model; and C gives the boundary layer thickness.

Figure 11a presents the plots of the pseudo–first–order and pseudo–second–order kinetic models for the adsorption of Cd(II) ions, and the parameters of the two kinetic models are given in Table 2. The results showed that, compared with the pseudo–first–order kinetic model, the pseudo-second-order kinetic model was more suitable for the experimental data for Fe_3_O_4_@SiO_2_@IIP and Fe_3_O_4_@SiO_2_@NIP in relation to Cd(II), and the correlation coefficients (*R*_2_^2^) were higher than 0.99. The pseudo-first-order kinetic model suggested that the adsorption process proceeded via diffusion of metal ions through the boundary layer on the adsorbent surface and that this adsorption process was controlled by the diffusion step, while the pseudo-second-order model assumed that the adsorption process was controlled by a chemisorption mechanism that involved electron sharing or electron transfer between the adsorbent and the adsorbate [36]. In general, the pseudo-second-order model could better explain the adsorption behavior because it was set across the whole range of the adsorption equilibrium time. Thus, it can be speculated that the adsorption of Cd(II) on Fe_3_O_4_@SiO_2_@IIP and Fe_3_O_4_@SiO_2_@NIP occurs via chemical adsorption.

The Weber–Morris model was introduced to further investigate the mechanism of the adsorption process. As shown in Figure 11b, the adsorption process could be divided into two stages by the Weber–Morris model. In the first stage, the diffusion rate constant *K*_1_ values for IIP and NIP were 3.83 and 2.81 mg·g^−1^·min^0.5^, respectively. This stage was attributed to the introduction of the template ions such that the IIP had more imprinted sites than the NIP, resulting in a higher diffusion rate constant for the IIP (*K*_IIP-1_) than for the NIP (*K*_NIP-1_) at this stage. In the second stage, both the IIP and NIP surfaces were almost saturated with adsorption after the first stage of rapid adsorption. This resulted in very low diffusion rate constants for IIP (*K*_IIP-2_) and NIP (*K*_NIP-2_) in the second stage of 0.39 and 0.44 mg·g^−1^·min^0.5^, respectively [37].

### 3.4. Adsorption Isotherms

Figure 12 shows the room temperature Cd(II) sorption isotherms for Fe_3_O_4_@SiO_2_@IIP and Fe_3_O_4_@SiO_2_@NIP (at 25 °C and pH = 6.0 with initial Cd(II) concentration in the range of 25–205 mg·L^−1^). At the corresponding initial concentrations, the adsorption capacity of Fe_3_O_4_@SiO_2_@NIP was clearly lower than that of Fe_3_O_4_@SiO_2_@IIP. This can be explained by the specific effect: the specific adsorption sites present on the surface of Fe_3_O_4_@SiO_2_@IIP were complementary to the template in terms of size and coordination geometries, which was favorable for the binding of Cd(II) with the recognition sites.

The Langmuir isotherm model, Freundlich isotherm model, and Scatchard model were fitted to the data to further investigate the adsorption mechanism of the adsorbent. The equations for the models are shown in Equations (9)–(11) [38]:(9)$$\frac{c_{e}}{q_{e}}=\frac{1}{q_{m}K_{L}}+\frac{c_{e}}{q_{m}}$$
(10)$$logq_{e}=logK_{F}+\frac{1}{n}logc_{e}$$
(11)$$\frac{q_{e}}{c_{e}}=\frac{q_{m}-q_{e}}{K_{s}}$$
where $c_{e}$ (mg·L^−1^) indicates the equilibrium concentrations of Cd(II) ions in solution, $q_{m}$ (mg·g^−1^) is the maximum adsorption capacity of the Cd(II) ion, $K_{L}$ (L g^−1^) is the Langmuir model constant, $K_{F}$ and *n* are the Freundlich model constants, and $K_{s}$ is the equilibrium dissociation constant at the binding sites in the Scatchard model.

The fitting results for the two models are shown in Figure 13, and the adsorption constants calculated from the corresponding isotherms with the correlation coefficients are given in Table 3. As shown in the table, the adsorption of Cd(II) onto Fe_3_O_4_@SiO_2_@IIP and Fe_3_O_4_@SiO_2_@NIP was well-fitted by the Langmuir model with a higher R^2^ (0.9988 and 0.9859). The consistency between the adsorption data and the Langmuir isotherm could be explained by the fact that the adsorption sites on the adsorbent surface were homogeneous, resulting in monolayer binding. Moreover, the Freundlich constant 1/*n* was less than 1, indicating a favorable process [11].

In the Scatchard model (Figure 14), the process of adsorption of Cd(II) on Fe_3_O_4_@SiO_2_@IIP exhibited two different linear relationships at different concentrations. This means that there were two types of Cd(II) binding sites on the Fe_3_O_4_@SiO_2_@IIP [39]. This was consistent with the results of the BET analysis. The first were non-specific binding sites with strong affinity formed by Cd(II) and two functional groups (-(OH)_2_PO and -COOH) in the functional monomer PBTCA, and the others were specific imprinted cavities formed on the surface of the imprinted polymer through the imprinting polymerization reaction. In contrast, for Fe_3_O_4_@SiO_2_@NIP, *q*_e_ showed a linear relationship with *q*_e_/*c*_e_, indicating that only non-specific binding sites existed on the Fe_3_O_4_@SiO_2_@NIP due to the fact that no template ion was introduced during the preparation of the Fe_3_O_4_@SiO_2_@NIP, resulting in an inability to generate imprinting cavities. Based on the intercept and slope of the fitted line, the dissociation constant (*K*_s_) and the maximum adsorption amount (*q*_m_) for the affinity site could be obtained, and the results are shown in Table 4; the smaller the *K*_s_, the higher the affinity is [40]. Notably, the fitted *K*_s_ values for each segment of the Fe_3_O_4_@SiO_2_@IIP (51.12 and 35.00) were smaller than that for the Fe_3_O_4_@SiO_2_@NIP (93.28), which proved that Fe_3_O_4_@SiO_2_@IIP had a stronger affinity for Cd(II) than Fe_3_O_4_@SiO_2_@NIP, and thus its adsorption capacity was higher than that of Fe_3_O_4_@SiO_2_@NIP.

### 3.5. Adsorption Thermodynamic

The effect of temperature on the adsorption equilibrium of Cd(II) for Fe_3_O_4_@SiO_2_@IIP and Fe_3_O_4_@SiO_2_@NIP (solution pH = 6.0, initial Cd(II) concentration of 50 mg·L^−1^, and temperature range of 25 °C–45 °C) was also investigated. Figure 15 shows the influence curve for temperature in relation to the adsorption capacity. The results showed that an increase in temperature was favorable for Cd(II) adsorption. Higher temperatures may increase the adsorbent pore size, allowing more Cd(II) to diffuse into the adsorbent. Furthermore, higher temperatures are more conducive to the diffusion of Cd(II) into the adsorbent system and its binding to the binding sites on the adsorbent surface.

To further explore the thermodynamic mechanism of the adsorption process, the Gibbs free energy equation (Equation (12)) and the Van’t Hoff equation (Equation (13)) were introduced as follows [41]:(12)$$∆G=-RT\ln\left(1000K_{c}\right)$$
(13)$$\ln K_{c}=-\frac{∆H}{RT}+\frac{∆S}{R}$$
where R is the universal gas constant (8.314 J mol^−1^ K^−1^), T is the absolute temperature (K), and $K_{c}$ is the equilibrium constant ($K_{c}=\frac{q_{e}}{c_{e}}$). The value of ∆G was calculated directly from the above relations, and the slope and the intercept of $lnK_{c}$ versus 1/T were used to calculate the values of ∆H and ∆S.

As shown in Table 5, the Δ*G* had negative values at different temperatures, indicating that the sorption of Cd(II) on Fe_3_O_4_@SiO_2_@IIP was spontaneous [40,42]. In addition, the ΔG decreased with the increase in temperature, indicating that a higher temperature was more favorable for the sorption process. On the other hand, the positive values for the ΔH suggested the endothermic nature of Cd(II) adsorption [43,44], which could have been due to some energy consumption for overcoming this potential barrier when the adsorption occurred near the imprinted adsorbent surface. It is worth noting that the ΔH value calculated in this work (8.56 kJ·mol^−1^) lay between the energy level of the electrostatic interactions of a Cd(II) complex (<4 kJ·mol^−1^) and that of a hydrogen bond (20.92 kJ·mol^−1^) [45]. When Fe_3_O_4_@SiO_2_@IIP reached adsorption saturation, increasing the temperature was favorable for its adsorption, and the increment in the maximum adsorption was related to the van der Waals force according to the value of the Δ*H*. For the Δ*S*, its positive value indicated that the higher temperature increased the solid–liquid interfacial irregularity of the adsorption process, while the relatively low entropy value indicated that the process maintained the relative regularity of Cd(II) on the polymer surface [44]. To conclude, the Cd(II) adsorption on the Fe_3_O_4_@SiO_2_@IIP surface was predominantly an endothermic, entropy-increasing process.

### 3.6. Selectivity Study and Reusability

In order to evaluate the selectivity of Fe_3_O_4_@SiO_2_@IIP and Fe_3_O_4_@SiO_2_@NIP for Cd(II), Zn(II), Ni(II), and Co(II) were selected as competing ions for selective adsorption experiments, and the results are shown in Figure 16 and Table 6. It can be seen that the adsorption selectivity of Fe_3_O_4_@SiO_2_@IIP for Cd(II) was much higher than that of the other metal ions, with a *D*_Cd(II)_ of 643.30 mL·g^−1^ and a *k*_Cd(II)/Co(II)_ of up to 16.58 for Fe_3_O_4_@SiO_2_@IIP. However, the *D*_Cd(II)_ value for Fe_3_O_4_@SiO_2_@NIP was smaller than those of *D*_Ni(II)_ and *D*_Co(II)_. This indicated that the phosphonate and carboxyl groups in the functional monomers had their own affinities, and the order of affinity for these ions was: Ni(II) > Co(II) > Cd(II) > Zn(II).

To explore the mechanism underlying this high selectivity, the binding energy of Cd(II) and other metals to PBTCA was calculated with the DFT method. As shown in Figure 17, the descending order of the binding energies of PBTCA with these metal ions was Ni(II), Co(II), Cd(II), and Zn(II), and the calculated results were consistent with the selective adsorption capacity of Fe_3_O_4_@SiO_2_@NIP. The results showed that Fe_3_O_4_@SiO_2_@IIP was made specific by the imprinting technology, and this specificity dominated the selective adsorption of Cd(II) from other interfering metal ions.

To investigate the stability and reusability of Fe_3_O_4_@SiO_2_@IIP, six adsorption–desorption experiments were carried out with 50 mg·L^−1^ Cd(II) solution using 2 mg·L^−1^ HCl aqueous solution as eluent, as shown in Figure 18. Compared to the initial cycle, the adsorption capacity of Fe_3_O_4_@SiO_2_@IIP only decreased by 7.32% for Cd(II), revealing the good regeneration capacity of Fe_3_O_4_@SiO_2_@IIP.

### 3.7. Selective Adsorption Mechanism

XPS analysis was carried out before and after the adsorption of Cd(II) by the imprinted polymer to further investigate the mechanism of adsorption of Cd(II) on Fe_3_O_4_@SiO_2_@IIP, and the XPS spectrum of the imprinted polymer after Cd(II) adsorption is shown in Figure 19a. The new characteristic peaks at 412.42 and 405.09 eV in the survey spectra were attributed to Cd 3d_3/2_ and Cd 3d_5/2_, respectively. The high-resolution XPS spectrum of Cd(II) is shown in Figure 19b, further indicating the imprinted polymer’s adsorption capacity for Cd(II). Deconvolution analysis of the O1s and N1s spectra was performed to examine the interactions between Cd(II) and the functional groups on Fe_3_O_4_@SiO_2_@IIP. Before Cd(II) adsorption, the O1s spectrum (Figure 19c) could be deconvoluted into two individual peaks with binding energies of 531.79 and 530.83 eV, which were ascribed to O atoms of the -COOH and -PO_3_H_2_ groups in the functional monomer PBTCA, respectively [46]. After Cd(II) adsorption, the O1s peaks shifted to higher binding energies of 532.23 and 531.03 eV, respectively. These shifts were caused by the empty orbitals of the cadmium atom sharing the electron cloud of the oxygen atom [19]. However, as shown in Figure 19d, the change in the binding energy of N1s was significantly less than 0.2 eV when Fe_3_O_4_@SiO_2_@IIP adsorbed Cd(II); therefore, the N atom was considered not to be involved in coordination. The results showed that the O atoms of the carboxylic acid groups and the phosphonic acid groups in PBTCA play a critical role in the high capture capacity for Cd(II).

The Fukui function calculated with DFT was used for further verification of the chelating sites of the PBTCA^5−^ molecule. Figure 20 shows the geometry optimization of PBTCA^5−^ and the Fukui functions (*f*^−^) of atoms on potential adsorption sites in PBTCA^5−^.

The results indicated that one O atom of the C-O group in the carboxyl group and two O atoms of the P-O groups in the phosphonate group in the PBTCA^5−^ molecule had high *f*^−^ values, and their electrons were more easily attacked by Cd(II), which constructed the specific binding sites and captured Cd(II) ions by chelation [47]. It should be noted that, although the third O atom in the phosphonate group also had a high *f^−^* value, it could not be coordinated with Cd(II) due to the steric effect. This result was consistent with the results of the XPS analysis.

The changes in bond lengths and Wiberg bond orders before and after adsorption also reflected the mechanism of Cd(II) adsorption on the imprinted polymer, and the results are listed in Table 7. The results showed that the bond lengths of the C-O bonds in the carboxyl group and the P-O bonds in the phosphonate group increased and the Wiberg bond orders were weakened after the adsorption of Cd(II). These changes were caused by the electron transfer from the O atoms of the carboxyl group and the phosphonate group to the vacant orbitals of Cd(II) following Cd(II) adsorption.

### 3.8. Comparison with Other Adsorbents for Cadmium Adsorption

The adsorption performance of Fe_3_O_4_@SiO_2_@IIP contrasted with that of other sorbents reported in the literature. As seen from Table 8, the Cd(II) adsorption ability of the Fe_3_O_4_@SiO_2_@IIP was superior or equivalent to other adsorbents with regard to equilibrium time and absorption capacity. Furthermore, the imprinted polymer prepared in this study could be easily recovered owing to the efficient and quick magnetic separation. Moreover, the ecofriendly and easy manufacturing methods, as well as the high chemical and thermal stability, will provide Fe_3_O_4_@SiO_2_@IIP with a wide range of applications in a variety of scientific fields.

## 4. Conclusions

A novel magnetic core-shell Cd(II) surface-imprinted polymer was synthesized using Cd(II) as a template, Fe_3_O_4_ as a matrix, MBA as a crosslinker, and PBTCA as a functional monomer by employing the surface imprinting technique and chemical grafting. The imprinted polymer was used to remove Cd(II) from aqueous solution, and the following conclusions were drawn. The introduction of carboxyl and phosphonic acid groups to the surface of Fe_3_O_4_@SiO_2_@IIP was confirmed. The pH value of the system played a crucial role in Cd(II) adsorption. The adsorption capacity increased with increases in pH value, with the maximum adsorption capacity reached at pH = 6.0. The kinetic adsorption process was described with pseudo-second kinetics as it was considered a chemisorption process. The Langmuir model fit the adsorption isotherm data best, implying a monolayer adsorption process. Thermodynamic studies demonstrated that the adsorption process was spontaneous, with an increase in entropy. Although PBTCA had low affinity for Cd(II), the imprinted polymer exhibited superior selectivity for Cd(II) compared to the non-imprinted polymer owing to the specificity obtained from the imprinting technique and dominated over the selective adsorption of Cd(II) from other interfering metal ions. Results from XPS and DFT calculations indicated that O atoms in the carboxyl and phosphonic acid groups in PBTCA were involved in the Cd(II) adsorption. Hence, Fe_3_O_4_@SiO_2_@IIP was proved to be a potential adsorbent for the rapid and selective adsorption and recovery of Cd(II) from aqueous solutions.

## Figures and Tables

**Figure 1 polymers-15-02416-f001:**
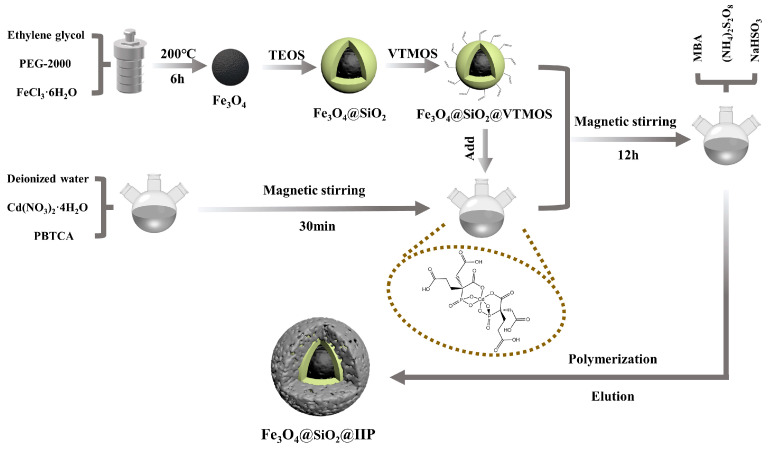
Schematic illustration of preparation of Cd(II)-IIP.

**Figure 2 polymers-15-02416-f002:**
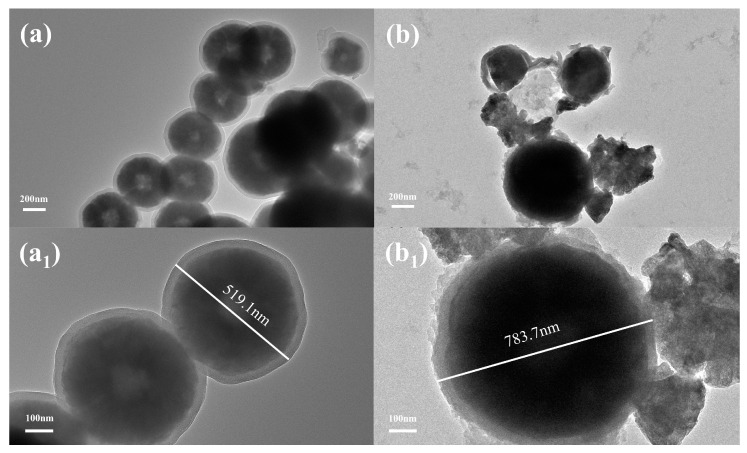
TEM images of (**a**) Fe_3_O_4_@SiO_2_ ((**a1**) enlarged TEM image of (**a**)) and (**b**) Fe_3_O_4_@SiO_2_@IIP ((**b1**) enlarged TEM image of (**b**)).

**Figure 3 polymers-15-02416-f003:**
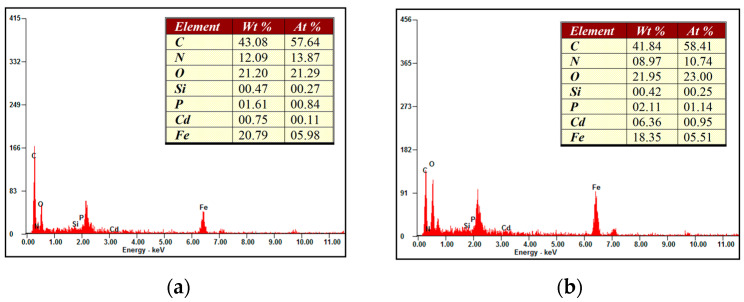
EDS spectra of (**a**) Fe_3_O_4_@SiO_2_@IIP before adsorption and (**b**) Fe_3_O_4_@SiO_2_@IIP after adsorption.

**Figure 4 polymers-15-02416-f004:**
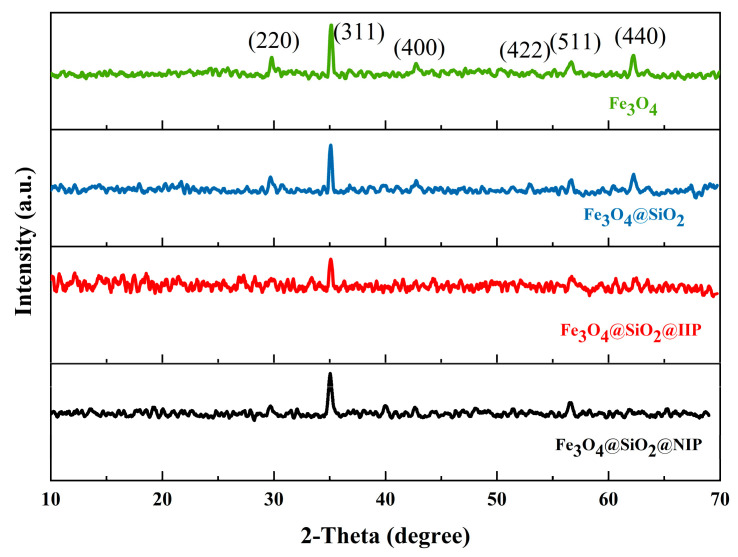
XRD patterns of Fe_3_O_4_, Fe_3_O_4_@SiO_2_, Fe_3_O_4_@SiO_2_@IIP, and Fe_3_O_4_@SiO_2_@NIP.

**Figure 5 polymers-15-02416-f005:**
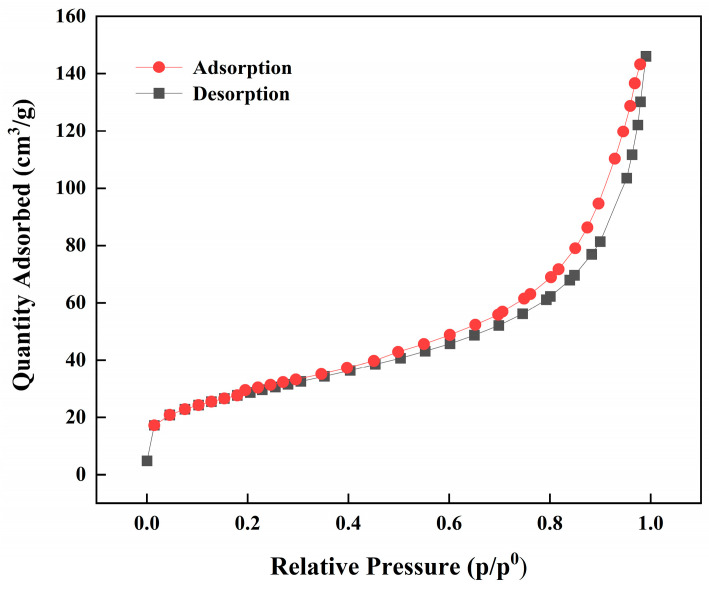
N_2_ adsorption–desorption isotherms for Fe_3_O_4_@SiO_2_@IIP.

**Figure 6 polymers-15-02416-f006:**
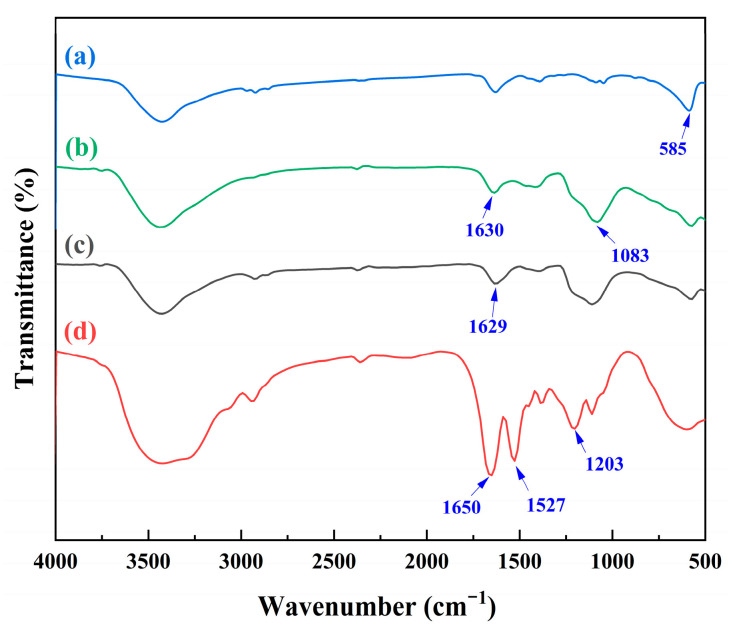
FTIR spectra of Fe_3_O_4_ (a), Fe_3_O_4_@SiO_2_ (b), Fe_3_O_4_@SiO_2_@VTMOS (c), and Fe_3_O_4_@SiO_2_@IIP (d).

**Figure 7 polymers-15-02416-f007:**
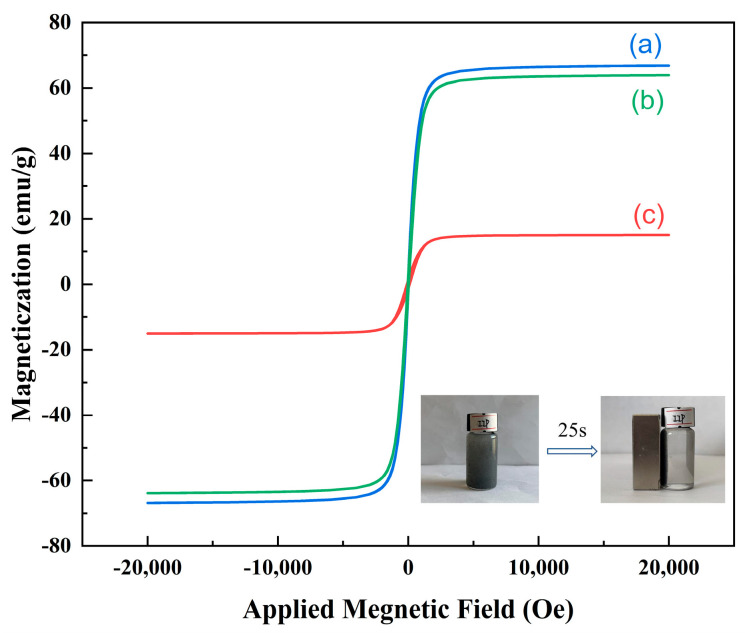
The hysteresis curves for Fe_3_O_4_ (a), Fe_3_O_4_@SiO_2_ (b), and Fe_3_O_4_@SiO_2_@IIP (c).

**Figure 8 polymers-15-02416-f008:**
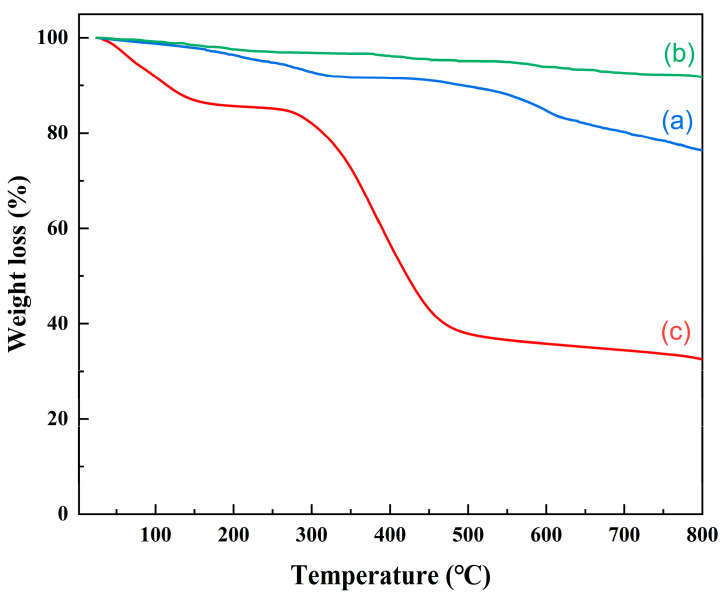
TG curves for Fe_3_O_4_ (a), Fe_3_O_4_@SiO_2_ (b), and Fe_3_O_4_@SiO_2_@IIP (c).

**Figure 9 polymers-15-02416-f009:**
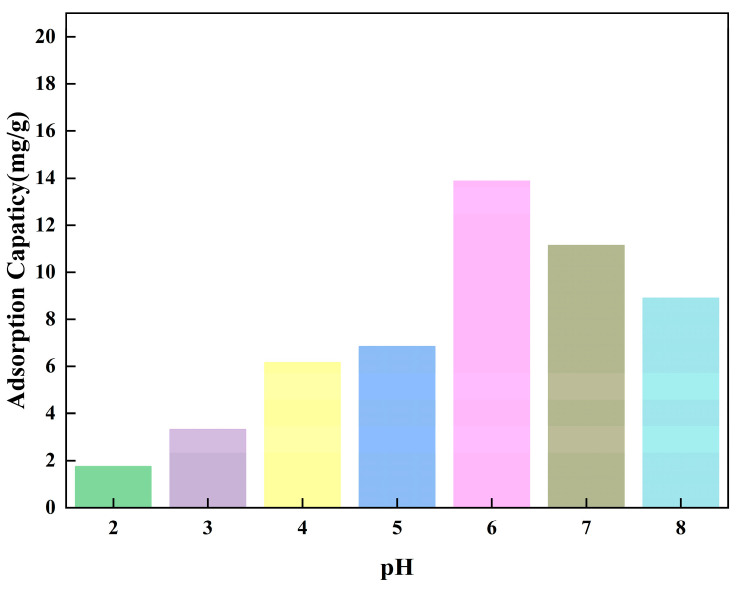
Effect of pH on the adsorption capacity of Fe_3_O_4_@SiO_2_@IIP.

**Figure 10 polymers-15-02416-f010:**
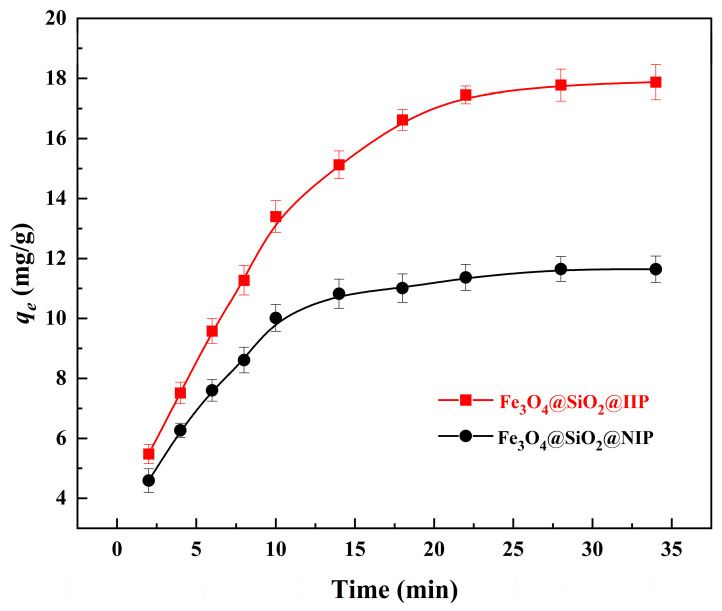
Effects of time on Cd^2+^ adsorption on Fe_3_O_4_@SiO_2_@IIP and Fe_3_O_4_@SiO_2_@NIP.

**Figure 11 polymers-15-02416-f011:**
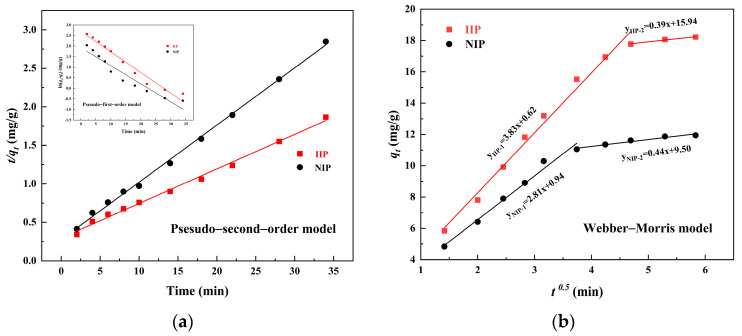
Pseudo–second–order kinetic model (the inset shows the pseudo–first–order kinetic model) (**a**) and Weber–Morris intra-particle diffusion model (**b**).

**Figure 12 polymers-15-02416-f012:**
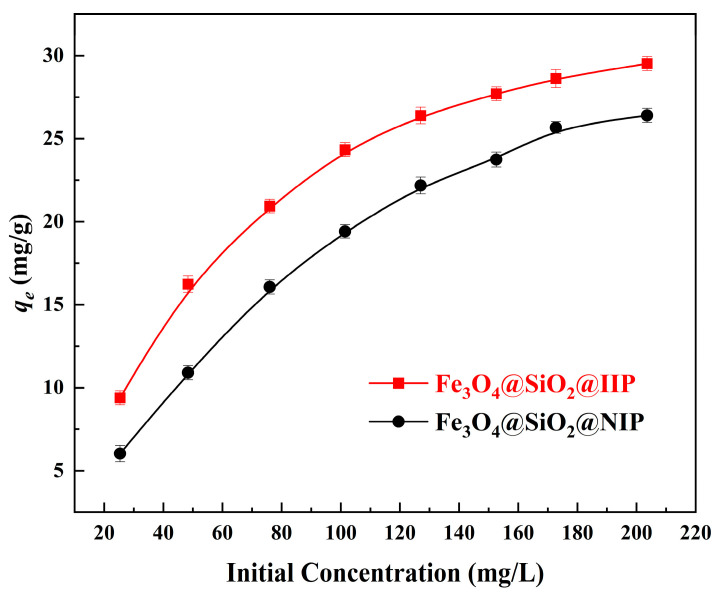
Effects of the initial concentration of Cd(II) on the adsorption capacity of Fe_3_O_4_@SiO_2_@IIP and Fe_3_O_4_@SiO_2_@NIP.

**Figure 13 polymers-15-02416-f013:**
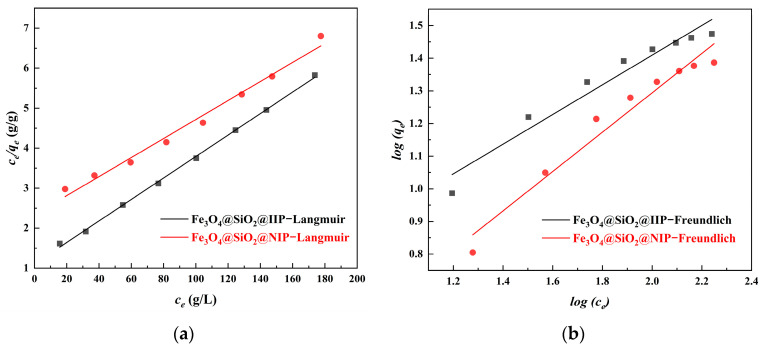
Isotherm model fitting curves for Fe_3_O_4_@SiO_2_@IIP and Fe_3_O_4_@SiO_2_@NIP. (**a**) Langmuir model and (**b**) Freundlich model.

**Figure 14 polymers-15-02416-f014:**
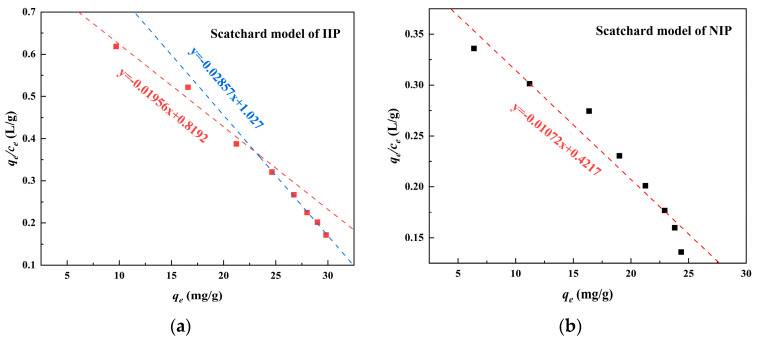
Scatchard model fitting for the IIP (**a**) and NIP (**b**).

**Figure 15 polymers-15-02416-f015:**
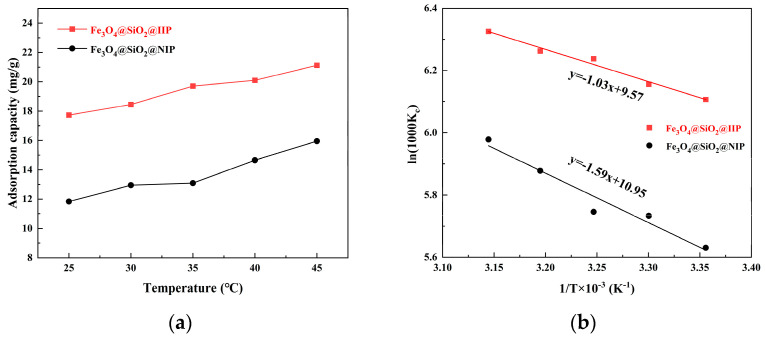
Effect of temperature on the adsorption (**a**) and plot of ln*K_c_* versus 1/*T* (**b**).

**Figure 16 polymers-15-02416-f016:**
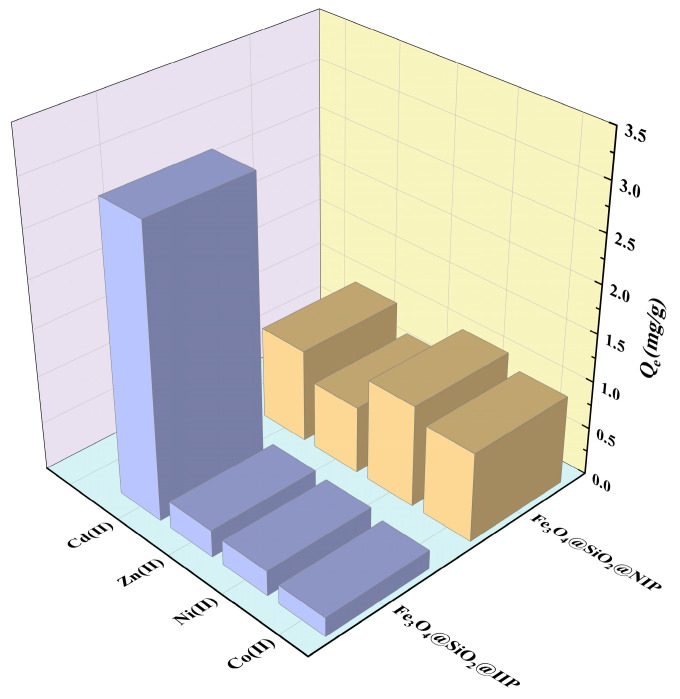
Selective adsorption of Fe_3_O_4_@SiO_2_@IIP and Fe_3_O_4_@SiO_2_@NIP.

**Figure 17 polymers-15-02416-f017:**
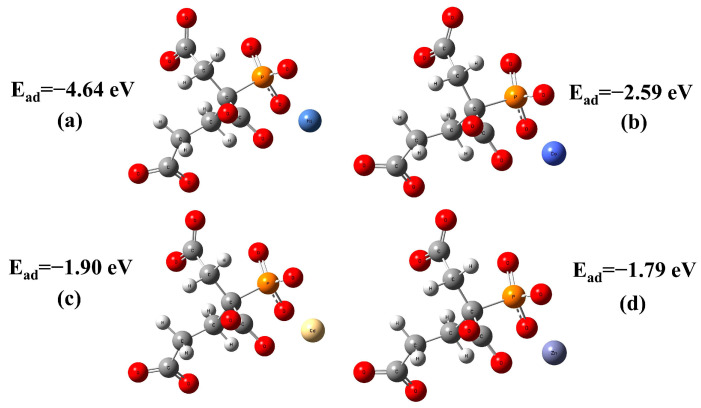
Binding energies of different ions to PTBCA: (**a**) PTBCA-Ni(II), (**b**) PTBCA-Co(II), (**c**) PTBCA-Cd(II), and (**d**) PTBCA-Zn(II).

**Figure 18 polymers-15-02416-f018:**
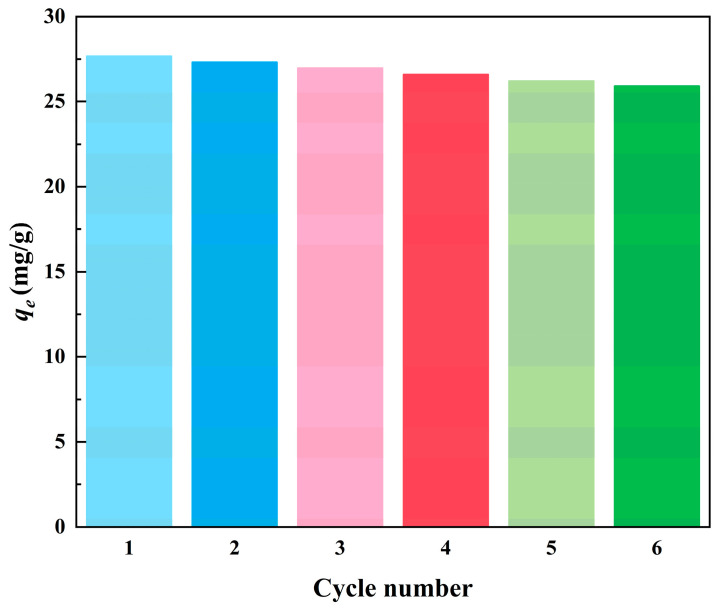
Regeneration performance of Fe_3_O_4_@SiO_2_@IIP in six adsorption-desorption cycles.

**Figure 19 polymers-15-02416-f019:**
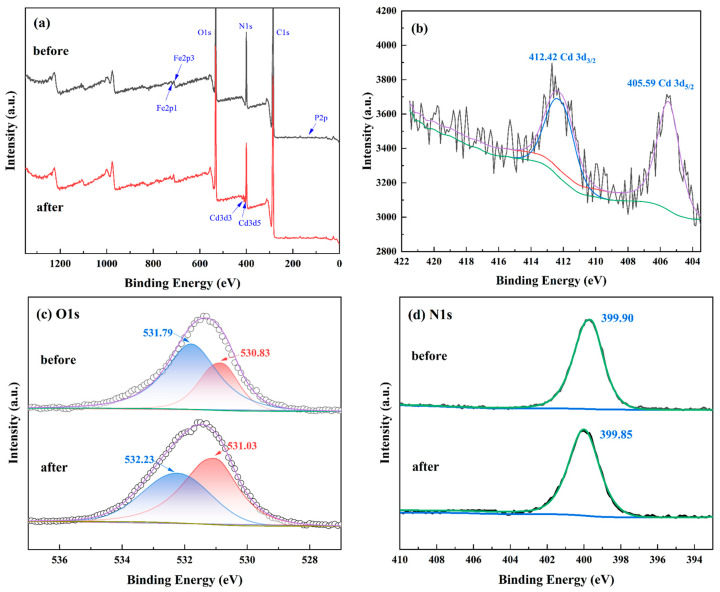
XPS spectra for IIP before and after Cd(II) adsorption (**a**), adsorption and XPS spectra for Cd3d (**b**), and high-resolution XPS spectra for O1s (**c**) and N1s (**d**) for IIP before and after adsorption of Cd(II).

**Figure 20 polymers-15-02416-f020:**
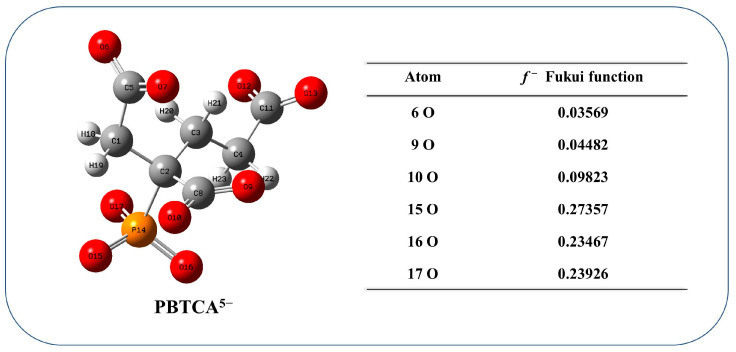
DFT calculations: geometric optimization of PBTCA and Fukui function (*f*^−^).

**Table 1 polymers-15-02416-t001:** Surface physical parameters of Fe_3_O_4_@SiO_2_@IIP and Fe_3_O_4_@SiO_2_@NIP.

Sample	Surface Area (m^2^·g^−1^)	Pore Volume (cm^3^·g^−1^)	Average Pore Diameter (nm)
Fe_3_O_4_@SiO_2_@IIP	107.990	0.235	8.675
Fe_3_O_4_@SiO_2_@NIP	79.455	0.182	8.747

**Table 2 polymers-15-02416-t002:** The adsorption kinetic parameters for IIP and NIP.

Adsorption	*q* _e, exp_	Pseudo-First-Order Kinetic Model			Pseudo-Second-Order Kinetic Model		
		*k* _1_	*q* _e, cal_	*R* _1_ ^2^	*k* _2_	*q* _e, cal_	*R* _2_ ^2^
IIP	18.21	0.2245	14.90	0.9658	0.0068	22.30	0.9945
NIP	11.94	0.1971	6.86	0.9264	0.0199	13.46	0.9978

**Table 3 polymers-15-02416-t003:** Isotherm model parameters used for the Fe_3_O_4_@SiO_2_@IIP and Fe_3_O_4_@SiO_2_@NIP.

Adsorbents	*q* _e, exp_	Langmuir			Freundlich		
		*K* _L_	*q* _m_	*R* ^2^	n	*K* _F_	*R* ^2^
Fe_3_O_4_@SiO_2_@IIP	29.82	0.02429	37.24	0.9988	2.199	3.162	0.9501
Fe_3_O_4_@SiO_2_@NIP	24.35	0.01018	42.03	0.9859	1.659	1.225	0.9592

**Table 4 polymers-15-02416-t004:** Parameters for the Scatchard model.

Adsorbents	*q*_e, exp_ (mg·g^−1^)	*K*_s_ (mg·L^−1^)	*q*_m_ (mg·g^−1^)	*R* ^2^
Fe_3_O_4_@SiO_2_@IIP	29.82	51.12	41.88	0.9583
		35.00	35.94	0.9966
Fe_3_O_4_@SiO_2_@NIP	24.35	93.28	39.34	0.9452

**Table 5 polymers-15-02416-t005:** Thermodynamic parameters for Cd(II) adsorption.

Adsorbent	Temperature (K)	Δ*G* (kJ·mol^−1^)	Δ*H* (kJ·mol^−1^)	Δ*S* (J·mol^−1^·K^−1^)
Fe_3_O_4_@SiO_2_@IIP	298	−15.13	8.56	79.56
	303	−15.51		
	308	−15.97		
	313	−16.03		
	318	−16.72		
Fe_3_O_4_@SiO_2_@NIP	298	−13.95	13.22	91.04
	303	−14.44		
	308	−14.71		
	313	−15.30		
	318	−15.81		

**Table 6 polymers-15-02416-t006:** Selectivity parameters for Fe_3_O_4_@SiO_2_@IIP and Fe_3_O_4_@SiO_2_@NIP.

Metal Ion	Fe_3_O_4_@SiO_2_@IIP			Fe_3_O_4_@SiO_2_@NIP			*k*′
	*q*_e_ (mg·g^−1^)	*D* (mL·g^−1^)	*k*	*q*_e_ (mg·g^−1^)	*D* (mL·g^−1^)	*k*	
Cd(II)	3.02	643.30		0.98	172.06		
Zn(II)	0.28	54.12	11.89	0.70	139.82	1.23	9.67
Ni(II)	0.26	51.78	12.42	1.07	230.34	0.75	16.56
Co(II)	0.20	38.81	16.58	0.93	197.39	0.87	19.06

**Table 7 polymers-15-02416-t007:** Partial bond lengths and Wiberg bond orders for PTBCA^5−^ and PTBCA-Cd(II).

Items	Bond Length (Å)			Wiberg Bond Order		
	C8-O10	P14-O16	P14-O17	C8-O10	P14-O16	P14-O17
PBTCA^5−^	1.262	1.530	1.532	2.394	2.136	2.112
PBTCA-Cd(II)^3−^	1.287	1.577	1.579	2.193	2.126	1.827

**Table 8 polymers-15-02416-t008:** Comparison of maximum adsorption capacities of Fe_3_O_4_@SiO_2_@IIP for Cd(II) with other adsorbents reported in the literature.

Adsorbent	Adsorption Capacity (mg·g^−1^)	Equilibrium Time (min)	Ref
Ion-imprinted polydopamine-coated magnetic graphene oxide	17.5	15	[48]
IIHP-VIN-MP	16.99	-	[49]
Vermiculite-600	8.50	120	[50]
MgO-ATP	24.50	180	[51]
Serpentine-700	17.68	120	[52]
C-ATP-SO_4_^2−^	13.29	120	[53]
C-ATP-Cl^−^	17.83	120	[53]
Fe_3_O_4_@SiO_2_@IIP	29.82	20	This work

## Data Availability

The data presented in this study are available on request from the corresponding author.

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
