# Peer review of "Selective Adsorption Behavior and Mechanism for Cd(II) in Aqueous Solution with a Recoverable Magnetie-Surface Ion-Imprinted Polymer"

_polymers, 2023, doi:10.3390/polym15112416_

Round 1

Reviewer 1 Report

This article needs major revision.

1.     Abstract: correct: secondary as second, and isothermal as isotherm. Check the whole manuscript for similar corrections.

2.     Adsorption Experiments must be written in more detail, and all the experimental conditions must be clear.

3.     XRD analysis of the materials must be included.  

4.     The effect of pH needs more explanation.

5.     Improve the Fig 9 (inset) quality.

6.      Adsorption kinetics need more explanation.

7.     Why did the authors use the linear Scatchard model while Langmuir and Freundlich fitted with the non-linear equation? Keep uniform.

8.     Similarly, kinetics is also fitted to linear equations. Fit with non-linear.

9.     DFT calculations are made by considering the chelation of Cd onto PBTCA. At the same time, adsorption is a process where physical adsorption and Cd(II) adsorption onto SiO2 and Fe3O4 may also occur. In this condition, fitting the DFT based on PBTCA chelation is corrected. Authors need to explain and justify.

10.  Authors must compare the adsorption capacity of Fe3O4@SiO2@IIP with pure Fe3O4@SiO2, Fe3O4, and SiO2. 

A general English editing and scientific explanation is needed. 

Author Response

Responds to the reviewer’s comments:

Referee: 1

Q1: Abstract: correct: secondary as second, and isothermal as isotherm. Check the whole manuscript for similar corrections.

Re: Thanks for your careful checks. We are sorry for our carelessness. Based on your comments, we have made the corrections to make the word harmonized within the whole manuscript, and marked it in red font.

Q2: Adsorption Experiments must be written in more detail, and all the experimental conditions must be clear.

Re: Thanks to the reviewers' comments. We have separately described the experimental conditions in detail when exploring each of the adsorption properties.

Q3: XRD analysis of the materials must be included.

Re: We thank the reviewer for the constructive comments. We have added XRD analyses to further strengthen our work (Line 159-170).

Q4: The effect of pH needs more explanation.

Re: Thank reviewer for the comment. We apologize that the pH explanation in the previous manuscript is not very clear and we have provided additional explanations in Lines 262-266.

Q5: Improve the Fig 9 (inset) quality.

Re: Thank you very much for your suggestion. We agree with the reviewer's comments that the situ picture of a is not clear and have replaced the image with new pictures.

Q6: Adsorption kinetics need more explanation.

Re: Thank you for your advice. We have provided additional explanations in Lines 296-301.

Q7: Why did the authors use the linear Scatchard model while Langmuir and Freundlich fitted with the non-linear equation? Keep uniform.

Re: We thank the reviewer for the constructive comments. We have re-fitting these models according to the reviewer's suggestion and keep uniform.

Q8: Similarly, kinetics is also fitted to linear equations. Fit with non-linear.

Re: Thank you for your recommend. To be consistent with the other fitted models, the kinetic model also used a linear fit.

Q9: DFT calculations are made by considering the chelation of Cd onto PBTCA. At the same time, adsorption is a process where physical adsorption and Cd(II) adsorption onto SiO2 and Fe3O4 may also occur. In this condition, fitting the DFT based on PBTCA chelation is corrected. Authors need to explain and justify.

Re: Thank you for asking. The magnetic imprinted polymer we have prepared is based on Fe304@SiO2, and a thin layer of imprinted polymer was polymerised on its surface by chemical grafting methods and imprinting techniques, therefore, the adsorption occurs on the thin layer of imprinted polymer and does not involve the adsorption of Fe3O4 and Fe3O4@SiO2. Also, according to question 10, we have done adsorption experiments with the two matrix materials mentioned above and the data results showed that they basically do not adsorb Cd(II). As for DFT calculations, we also introduced the keyword of dispersion energy correction in DFT calculations, aiming to take into account the role of some physical adsorption.

Q10: Authors must compare the adsorption capacity of Fe3O4@SiO2@IIP with pure Fe3O4@SiO2, Fe3O4, and SiO2.

Re: Thank reviewer for the comment. We conducted adsorption experiments under the conditions of initial Cd(II) concentration of 50 mg L-1, pH=6.0 and temperature of 25°C to compare the adsorption capacity of Fe3O4@SiO2@IIP with pure Fe3O4@SiO2, Fe3O4. The results are listed in Table.

Adsorbents

Adsorption capacities (mg·g-1)

Fe3O4

0.79

Fe3O4@SiO2

1.22

Fe3O4@SiO2@IIP

23.40

Special thanks to you for your good comments.

We tried our best to improve the manuscript and made some changes in the manuscript. These changes will not influence the content and framework of the paper. And here we did not list the changes but marked in red in revised manuscript.

Reviewer 2 Report

The manuscript shows the modification of Fe3O4@SiO2 for adsorption of cadmium for the application of impurities removal in water. The manuscript shows promising results for removing cadmium particles from water, and this manuscript should be accepted after revision is performed.

1. The scale in Figure 2 cannot be read clearly. Please adjust the size of images.

2. Under thermal stability analysis, the authors observed a decrease in mass loss from 23.5% to 8.22% and explained that this is part of successful coating. Does that mean SiOhas no contribution towards mass loss?

3. In Figure 6, materials (a) and (b) are also observed in experiencing first drop at 150oC, but not significant. Please revise the paragraph.

4. Line 275 - 277, could the material adsorption be approaching equilibrium? This seems like NIP has lower uptake due to lower surface area and reaches equilibrium earlier as compared to IIP. Also, BET surface analysis was not performed to elaborate the surface characteristics of the materials.

5. Please add error bars in Figure 11.

6. Line 321-323, this can be explained with N2 adsorption isotherm experiment. It is essential to have such results included in the manuscript.

7. Line 347, given the hypothesis from the that such process is endothermic, this process is likely a chemisorption process. Thus, the explanation with van der Waals forces do not make sense as this work is a chemisorption process. On the other hand, enthalpy is relatively in the range for physical adsorption, which is supposed to be an exothermic process. I strongly advise the authors to revise their results and explain this phenomenon. 

Line 417-418, the authors mentioned about chelation behind the adsorption of Cd(II). However, this does not correlate with endothermic reaction for the adsorption process. Please provide sufficient evidence and discussion about the hypothesis.

There are some minor typos in the manuscript, which can be seen at line 38, 45 (space before citation bracket) and 144 (subscript of 1 in a1)

Author Response

Responds to the reviewer’s comments:

Referee: 2

Q1: The scale in Figure 2 cannot be read clearly. Please adjust the size of images.

Re: Thanks for your feedback. We have re-adjusted in the revised manuscript.

Q2: Under thermal stability analysis, the authors observed a decrease in mass loss from 23.5% to 8.22% and explained that this is part of successful coating. Does that mean SiO2 has no contribution towards mass loss?

Re: Thank you very much for your suggestions. Fe3O4 and Fe3O4@SiO2 experienced a weight loss of 23.5% and 8.22%, respectively, as the temperature increased from 25 °C to 800 °C. This is attributed to the successful coating of SiO2 on the surface of Fe3O4. The modification of SiO2 resulted in the improvement of the thermal stability.

Q3: In Figure 6, materials (a) and (b) are also observed in experiencing first drop at 150 ℃, but not significant. Please revise the paragraph.

Re: We thank the reviewer for the constructive comments. We have re-written this part according to the reviewer's suggestion.

Q4: Line 275 - 277, could the material adsorption be approaching equilibrium? This seems like NIP has lower uptake due to lower surface area and reaches equilibrium earlier as compared to IIP. Also, BET surface analysis was not performed to elaborate the surface characteristics of the materials.

Re: Thanks for your great suggestion on improving the accessibility of our manuscript. Line 275 – 277, the material is close to equilibrium. Also, we have added BET test analyses to further refine the study (in Line 172-186).

Q5: Please add error bars in Figure 11.

Re: Thank you for your recommend. We re-fitted the adsorption isotherm with a linear Langmuir and Freundlich model. Error bars have been added to figures 8 and 10.

Q6: Line 321-323, this can be explained with N2 adsorption isotherm experiment. It is essential to have such results included in the manuscript.

Re: According to the reviewers’ comments, we have made modifications to our manuscript and supplemented extra data to make our results convincing.

Q7: Line 347, given the hypothesis from the that such process is endothermic, this process is likely a chemisorption process. Thus, the explanation with van der Waals forces do not make sense as this work is a chemisorption process. On the other hand, enthalpy is relatively in the range for physical adsorption, which is supposed to be an exothermic process. I strongly advise the authors to revise their results and explain this phenomenon. Line 417-418, the authors mentioned about chelation behind the adsorption of Cd(II). However, this does not correlate with endothermic reaction for the adsorption process. Please provide sufficient evidence and discussion about the hypothesis.

Re: Thanks to the reviewer's suggestion. We apologize that the thermodynamic explanation of adsorption in the previous manuscript is not very clear. In the study of adsorption thermodynamics, we aimed to investigate the effect of temperature on their maximum adsorption capacity, and when IIP reach adsorption saturation, the continued increase in temperature was favorable for their adsorption, and the increment that occurred in the case of maximum adsorption capacity was derived by ΔH is related to van der Waals forces, which do not conflict with the chelation of Cd(II) with the imprinted sites in PBTCA, and this chelation process occurs during the adsorption process. Again, we apologize for the inaccurate presentation in the previous manuscript, and we have rewritten the relevant content in Lines 393-396.

Special thanks to you for your good comments.

We tried our best to improve the manuscript and made some changes in the manuscript. These changes will not influence the content and framework of the paper. And here we did not list the changes but marked in red in revised manuscript.

Round 2

Reviewer 1 Report

acceptable

revision of minor editing 

Author Response

Dear reviewer:

    Special thanks to you for your good comments.  The language of the manuscript has also been polished, especially the Abstract, Introduction and Conclusion have been comprehensively revised, and here we did not list the changes but marked in red in revised manuscript.

We appreciate Reviewers’ warm work earnestly, and hope that the correction will meet with approval.

Once again, thank you very much for your comments and suggestions.

Reviewer 2 Report

The revision is acceptable, but please add legends in Figure 5 to show which lines represent adsorption and desorption.

Minor English editing is needed. 

Author Response

Dear reviewer:

    Special thanks to you for your good comments.  The legends in Figure 5 to show which lines represent adsorption and desorption was added. In additon, the language of the manuscript has also been polished, especially the Abstract, Introduction and Conclusion have been comprehensively revised, and here we did not list the changes but marked in red in revised manuscript.

We appreciate Reviewers’ warm work earnestly, and hope that the correction will meet with approval.

Once again, thank you very much for your comments and suggestions.
